# Sleep Deprivation in the Forward-Forward Algorithm

**Mircea-Tudor Lică & David Dinucu-Jianu**
Department of Computer Science
Delft University of Technology
Delft, Netherlands
`{M.Lica,D.Dinucu-Jianu}@student.tudelft.nl`

## Abstract

This paper aims to explore the separation of the two forward passes in the Forward-Forward algorithm from a biological perspective in the context of sleep. We show the size of the gap between the sleep and awake phase influences the learning capabilities of the algorithm and highlight the importance of negative data in diminishing the devastating effects of sleep deprivation.

## 1 Introduction

The Forward-Forward (FF) algorithm (Hinton, 2022) introduces a new learning procedure that provides a feasible model of how learning works inside the cortex. In contrast with backpropagation (Rumelhart et al., 1986), which has been previously shown to be an implausible explanation for learning in the brain (Lillicrap et al., 2020), the Forward-Forward algorithm aims to avoid the large memory footprint and overhead computation arising from the backward pass by introducing two separate forward passes that optimize opposite objectives. During training, one forward pass operates on real or positive data, while the other uses negative data, which can be generated internally by the network through top-down connections or supplied externally.

One of the questions posed in the original paper was concerned with the possible effects the separation of the two forward passes might have on the capabilities of the algorithm. If proven successful, this separation would allow for energy-efficient applications in which real data would be processed continuously while the negative data would be generated and used at a separate point in time. Moreover, by isolating the forward passes, the Forward-Forward algorithm could potentially be implemented in the brain since positive data would be processed during the awake phase and negative data during a subsequent sleep phase. This separation is, however, briefly explored, and there was no comprehensive study on the effects it could have on the performance of the algorithm.

The aim of this paper is to further explore the separation of the two forward passes and examine the performance on several image datasets. We will specifically concentrate on the scenario where the awake phase is longer than the sleep phase and investigate the impact of sleep deprivation on the learning capabilities of the Forward-Forward algorithm. The code is available at https://github.com/mirceatlx/FF.

## 2 Sleep

According to Crick (Crick & Mitchison, 1983), the function of rapid eye movement (REM) sleep (Siegel, 2005) is to remove unwanted modes of behaviour that arise from either the expansion of the brain or through experience. Similar to Hopfield Networks (Hopfield, 1982), where trying to store an excessive amount of patterns leads to forgetting and the creation of spurious memories, overloading the brain with new experiences leads to certain parasitic modes of behaviour. We believe that this connection between experience and sleep encapsulates the relationship between the two forward passes in the Forward-Forward algorithm. Thus, the forward pass that uses positive data represents new experiences or information, while the second forward pass makes use of negative data to suppress parasitic or redundant patterns created by the exposure to real data. Since we

present sleep as one or more forward passes running with negative data, we define sleep deprivation as learning with positive data (awake phase) for prolonged periods of time followed by a short burst of sleep.

The forward pass that corresponds to the sleep phase is characterized by the objective it tries to optimize and the choice of negative data (see Appendix B). While the optimization objective is just the opposite of the objective used in the positive pass, there are multiple possibilities to represent negative data. In an ideal scenario, the negative data would be generated from the top-down connections of the model, similar to Predictive Coding (Millidge et al., 2021). However, it is possible to provide this data externally using some sort of generation strategy. The paper that introduces the FF algorithm provides some possible techniques, such as inserting a wrong label in real data or creating hybrid images by combining two samples from the dataset (see Appendix A).

## 3 EXPERIMENTS AND RESULTS

We present the possible separation of the positive and negative forward passes using the MNIST for handwritten digits and Fashion-MNIST (Xiao et al., 2017) datasets by utilizing the two negative data generation strategies described in section A. Table 1 highlights the scenario where the model spends an equal amount of time in the awake and sleep phase. More specifically, we separate the two forward passes by alternating between equal periods spent in both stages, where a period is defined as a fixed number of batches of either real or negative data.

Table 1: Accuracy of models with equal awake and sleep phases, starting from 1 batch per phase up to 128. The model architecture and training procedure are described in section C.

| Dataset | Negative data | 1 | 2 | 4 | 8 | 16 | 32 | 64 | 128 |
|---------|---------------|-----|-----|-----|-----|-----|-----|-----|-----|
| MNIST | Wrong label | 89% | 88% | 81% | 63% | 35% | 11% | 10% | 9% |
| MNIST | Masks | 88% | 84% | 85% | 78% | 74% | 49% | 23% | 11% |
| Fashion-MNIST | Wrong label | 73% | 63% | 59% | 54% | 20% | 14% | 10% | 10% |
| Fashion-MNIST | Masks | 56% | 58% | 53% | 50% | 45% | 36% | 30% | 22% |

On the other hand, Table 2 features a sleep deprivation scenario where there is only one batch of negative data followed by multiple batches of positive data.

Table 2: Accuracy of models using unequal phases. The period of the positive data ranges from 1 to 16, and the negative phase is fixed at 1. The empty lines in the table represent models that experienced no learning.

| Dataset | Negative data | 1 | 2 | 4 | 8 | 16 |
|---------|---------------|-----|-----|-----|-----|-----|
| MNIST | Wrong label | 89% | 10% | 9% | - | - |
| MNIST | Masks | 88% | 78% | 75% | 73% | 10% |
| Fashion-MNIST | Wrong label | 73% | 56% | 47% | 14% | 10% |
| Fashion-MNIST | Masks | 56% | 55% | 54% | 52% | 49% |

We find that the accuracy of the algorithm degrades as the number of batches in the awake and sleep phases increases. Using masks for generating negative data greatly improves the performance of both sleep deprivation and equal awake-sleep scenarios and allows for longer awake periods.

## 4 CONCLUSION

Our research reveals the relationship between the separation of the two forward passes and the dynamics between sleep and awake phases found in the brain, demonstrating a strong correlation between the learning performance and the generation strategy for negative data, particularly under conditions mimicking sleep deprivation. Better approaches for generating negative data allow the model to stay awake for longer periods of time, enhancing its possible usage in energy-efficient on-chip learning. Future research in this area should be conducted in order to fully explore the sleep properties of the algorithm from both a practical and biological perspective.

URM Statement

The authors acknowledge that at least one key author of this work meets the URM criteria of ICLR 2023 Tiny Papers Track.

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

## A  NEGATIVE DATA

We employed the use of two different negative data generation strategies. Our first approach involved setting a wrong label on a train data point and using this as negative data; this can be seen in Figure 1. We later experimented with using a mixture of two data points combined using a predefined mask as presented in the Forward-Forward paper (Hinton, 2022); this approach can be visualized in Figure 2. The latter approach allowed our model to perform significantly better than the former, allowing us to separate the awake and sleep phases more.

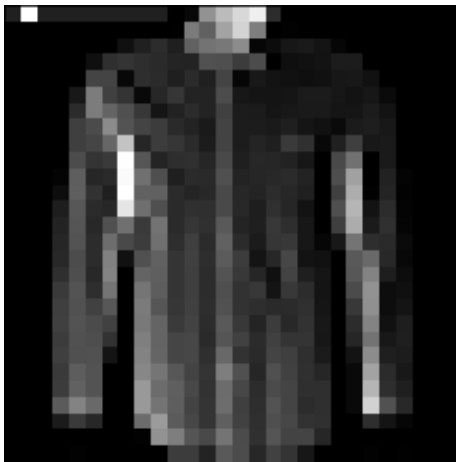

Figure 1: Fashion-MNIST image showing a label overlayed on top of the image. The first ten pixels of the image are set to 0 and the label indicating the class is set to 1.

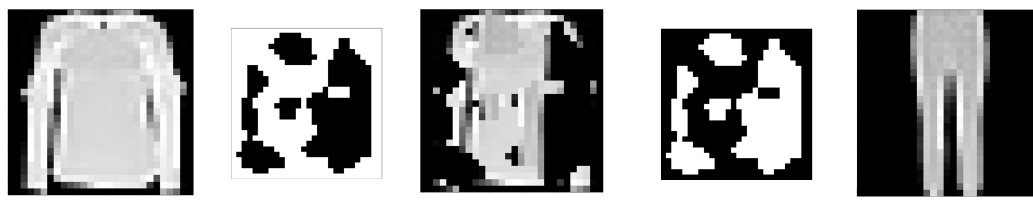

Figure 2: Negative data generation with masks on the Fashion-MNIST dataset. The image in the middle is the result of combining the lateral images with the generated mask and its inverse.

## B  LOSS CALCULATION

The loss calculation employed in our research is inspired by an open-source rendition of the algorithm delineated by Hinton [1]. While the original approach, as presented in the cited paper, utilizes the logistic function in loss computation, we have elected to use the Softplus function due to its superior empirical performance.

The Softplus function is mathematically defined as:

$$\text{Softplus}(x) = \log(1 + e^x) \tag{1}$$

---

[1]https://github.com/mohammadpz/pytorch_forward_forward

As a part of our algorithm, we compute two distinct loss values corresponding to positive and negative data samples. The concept of a 'threshold' is introduced, a hyperparameter that we have set to 1.5 in the context of this study, and represents the needed goodness for a layer to be considered activated.

$$\text{Loss}_{\text{pos}} = \text{Softplus}(-\text{Goodness}_{\text{pos}} + \text{Threshold}) \tag{2}$$

$$\text{Loss}_{\text{neg}} = \text{Softplus}(\text{Goodness}_{\text{neg}} - \text{Threshold}) \tag{3}$$

Both loss types aim to optimize different objectives; the loss for positive samples works to maximize layer goodness, whereas the loss for negative data strives to minimize it. The algorithm effectively differentiates between positive and negative data through the alternate or combined application of these losses.

Our approach stands in contrast to the original algorithm, where both loss components were averaged to calculate the total loss for parameter updates. We have diverged from this by incorporating each loss component independently during the updating process, allowing for a separation of the phases of the algorithm.

## C  MODEL ARCHITECTURE AND TRAINING PROCESS

All experiments are conducted with the same architecture consisting of an input layer of 784 and three fully connected hidden layers, all having 500 neurons. Each layer is trained for 50 epochs with a threshold of 1.5. We make use of the Adam optimizer (Kingma & Ba, 2015) with a batch size of 512. The learning rate is set at 0.001 for the negative phase, and the positive phase learning rate is calculated as follows.

$$\text{positive\_lr} = \frac{0.001}{\text{awake\_period}} \tag{4}$$

Where *awake_period* represents the number of batches of positive data the model will train on before switching to negative data.

While training with a substantial quantity of batches during the positive phase, it is possible to encounter situations where an epoch concluded before reaching the stipulated number of batches. In such instances, the training sequence continues into the next epoch, picking up right from where it left off in the previous one.

## D  PERFORMANCE OF UNSEPARATED PHASES

The initial implementation described by Hinton (2022) with unseparated positive and negative phases shows the best results on all datasets and training strategies. The results of our reimplementation of the algorithm can be found in Table 3.

Table 3: Test accuracy of the Forward Forward algorithm with unseparated phases.

| Dataset | Negative data | Unseparated |
|---|---|---|
| MNIST | Wrong label | 97.7% |
| MNIST | Masks | 96% |
| Fashion-MNIST | Wrong label | 88.7% |
| Fashion-MNIST | Masks | 86% |

