# OpenReview forum: "Sleep Deprivation in the Forward-Forward Algorithm"
_ICLR.cc/2023/TinyPapers — Submitted to Tiny Papers @ ICLR 2023_

### Official Review · Reviewer_s4HS · 2023-03-20

**Confidence:** 3

**Summary Of Contributions:**

The authors present the relationship between sleep patterns (monophasic sleep and sleep deprivation) and the learning performance of the Forward-Forward algorithm, a new (2022) learning procedure that consists of two forward passes, one with positive data and one with negative data (in the following as sleep). There seems to be a correlation between negative data (both in quantity and quality) and learning performance.

**Rating:**

Clear, Correct, and Reproducible (CCR): a submission which meets the reviewing criteria

**Strengths And Weaknesses:**

### Strengths:
1. The paper is well-written and easy to follow.
2. The claims and conclusions are justified by the findings

**Suggested Changes:**

The authors provide interesting findings between sleeping patterns and learning performance, seen in Tables 1 and 2. Specifically, it can be seen that using masks as negative data offer a significant advantage concerning model accuracy.
For future research, it would interesting to investigate different types of masks on more complicated datasets (a starting point could be CIFAR-10, like in the Forward-Forward paper).

---

> ### Author Response · Authors · 2023-05-31
> **Revision update**
>
> Dear ICLR TinyPaper Reviewer,
>
> Thank you for taking the time to review our paper and provide valuable feedback. We appreciate your insightful comments and suggestions.
>
> Regarding future investigations, we briefly tested the separation of the two forward passes on the CIFAR-10 dataset. However, the results were inconclusive so we decided to not include them in the revised version. We encourage more research in this direction.
>
> Best regards,
> Mircea Lica and David Dinucu-Jianu

---

### Author Response · Authors · 2023-05-31
**Opt-in for archival**

Dear reviewers and ICLR Chairs,

Thank you for your feedback and comments on our paper. We appreciate the opportunity to engage in this dialogue and receive valuable insights from experts in the field.

In response to the question of whether we wish to opt-in for archival, we would like to confirm our decision to opt-in.

Best regards,
Mircea Lica and David Dinucu-Jianu

---

### Comment · Area_Chair_jvAD · 2023-06-08
**Archival Criterion Check**

This work meets the threshold for archival, contains the URM statement, and is deanonymized.

---

### Meta-Review · Area_Chair_jvAD · 2023-04-08

**Recommendation:** Invite to revise
**Confidence:** 2

**Metareview:**

The reviewer rates this paper as CCR. I was interested in the subject of the paper so I read it but was left confused. The lack of clarity early in the paper makes it difficult to assess the conclusions. Notwithstanding my confusion, it appears to me that the findings might be of interest if the paper is revised to be much more clearly written.

**Summary:**

The paper explores the relative importance of the two phases of Hinton's Forward-Forward algorithm. Networks are trained with FF on MNIST and Fashion-MNIST. The paper is not very clear.

**Comments And Feedback To The Authors:**

The description of the FF algorithm is poor and confusing. Section 2 (Sleep) is entirely cryptic. Many terms and objectives are not defined or not well-motivated. For example, what is "positive/negative" data? What is "goodness" of a layer? What is REM sleep and what is its function? What is "coupling?" What is the "hypothesis" of the paper? Although Section 3 starts with the sentence "To test the hypothesis presented in this paper" I can't find any such hypothesis earlier in the paper.

**Reason For Not Giving A Higher Recommendation:**

The paper needs substantial clarification on the motivation, terminology, and main objective. Due to lack of clarity, currently it is difficult to assess the correctness or reproducibility of the paper.

**Reason For Not Giving A Lower Recommendation:**

N/A

---

> ### Author Response · Authors · 2023-05-31
> **Revision update**
>
> Dear ICLR TinyPaper Reviewer,
>
> Thank you for taking the time to review our paper and provide valuable feedback. We appreciate your insightful comments and suggestions.
>
>
> Based on the feedback we received, we decided to revise several sections of the paper. We quickly realised that the constraints imposed by the TinyPaper track (2 pages max.) limit our ability to thoroughly explain all the concepts we introduced in the original version. Thus, we decided to concentrate on the most important ideas we want to present, namely the separation of the two forward passes and sleep deprivation (this is explains the reason we have a new title in the revised version).
>
> Most of the feedback received was concerned with the lack of clarity in motivation and main objective. Moreover, some of the terminology was explained poorly or not at all.  We believe the revised version solves these problems with the following modifications:
>
> - There is a new explanation of the Forward-Forward algorithm that only concentrates on the relevant information for our paper.
> - The motivation and main objective is stated in a clear way such that it is easy for the reader to quickly understand what the paper is about
> - All the terminology we introduce clearly defined either in the main text or in the appendix. We tried to design the text such that the reader can understand the main idea without visiting the Appendix  section
> - Section 2 of the paper is completely revised, we now explain the relation between FF and sleep in a clearer way. We link some of the concepts with other techniques such as Predictive Coding and Hopfield Networks. Moreover, we clearly define what we consider sleep deprivation, awake-sleep phases and positive/negative data.
> - We removed redundant information that is not relevant to the subject of the paper.
>
>
> To provide the more information about our method and experiments, we added new sections in the Appendix that aim to further clarify some aspects of the paper.
>
> Lastly, in order to allow for the reproducibility and correctness of our results, we created a public Github repository that contains the code we used during our experiments. The link  to the repository is added in the Introduction section.
>
>
> Best regards, Mircea Lica and David Dinucu-Jianu

---

### Decision · Program_Chairs · 2023-04-09

Revision accepted; invite to archive